# Developmental Temperature Affects Life-History Traits and Heat Tolerance in the Aphid Parasitoid *Aphidius colemani*

**DOI:** 10.3390/insects12100852

**Published:** 2021-09-22

**Authors:** Mey Jerbi-Elayed, Vincent Foray, Kévin Tougeron, Kaouthar Grissa-Lebdi, Thierry Hance

**Affiliations:** 1Earth and Life Institute, Ecology and Biodiversity, Université Catholique de Louvain, 1348 Louvain-la-Neuve, Belgium; kevin.tougeron@uclouvain.be (K.T.); thierry.hance@uclouvain.be (T.H.); 2Institut de Recherche sur la Biologie de l’Insecte, UMR 7261, CNRS, Université de Tours, 37200 Tours, France; vincent.foray@univ-tours.fr; 3EDYSAN, UMR CNRS 7058 (Écologie et Dynamique des Systèmes Anthropisés), Université de Picardie Jules Verne, 80090 Amiens, France; 4Department of Plant Protection, Institut Agronomique de Tunisie, Université de Carthage, Carthage 28327, Tunisia; kaogrissa@gmail.com

**Keywords:** phenotypic plasticity, temperature, critical thermal limits, fat reserves, biological control

## Abstract

**Simple Summary:**

Individual insects can experience different temperatures across their lives. In consequence, development of immature stages can take place at temperatures contrasting with those that the adult will encounter. In order to study the adaptation of organisms to warm environments, it is interesting to establish the links between developmental temperature and responses of the adult generation, in terms of physiology, morphology, and behavior. We exposed the parasitoid *Aphidius colemani* to three temperatures during larval development (10, 20, or 28 °C). We then measured the responses of adult parasitoids regarding their traits (physiology, morphology) and their tolerance to high temperatures. Mass, lipid reserves, survival, and heat tolerance of adults were affected by the rearing temperature regime. This means that, overall, developmental temperatures may affect how insects respond to a given temperature once they are adults, which may consequently influence the ability of parasitoids to control their aphid hosts. These results are important in the context of biological control in hot environments, such as the Maghreb, where this species of parasitoid is used in mass release techniques to control aphids in various crops.

**Abstract:**

Developmental temperature plays important roles in the expression of insect traits through thermal developmental plasticity. We exposed the aphid parasitoid *Aphidius colemani* to different temperature regimes (10, 20, or 28 °C) throughout larval development and studied the expression of morphological and physiological traits indicator of fitness and heat tolerance in the adult. We showed that the mass decreased and the surface to volume ratio of parasitoids increased with the development temperature. Water content was not affected by rearing temperature, but parasitoids accumulated more lipids when reared at 20 °C. Egg content was not affected by developmental temperature, but adult survival was better for parasitoids reared at 20 °C. Finally, parasitoids developed at 20 °C showed the highest heat stupor threshold, whereas parasitoids developed at 28 °C showed the highest heat coma threshold (better heat tolerance CTmax1 and CTmax2, respectively), therefore only partly supporting the beneficial acclimation hypothesis. From a fundamental point of view, our study highlights the role of thermal plasticity (adaptive or not) on the expression of different life history traits in insects and the possible correlations that exist between these traits. From an applied perspective, these results are important in the context of biological control through mass release techniques of parasitoids in hot environments.

## 1. Introduction

Temperature is a major abiotic determinant of the fitness of poikilothermic organisms, such as insects. It affects the main physiological and developmental functions, immunity, and metabolic rates [1,2], as well as several morphological and behavioral life-history traits, such as locomotion, foraging, and mating [3,4,5]. For example, an extended longevity is observed for several biological agents reared at intermediate temperature regimes [6,7]. As a textbook case, the temperature-size rule states that ectotherms grow larger at lower temperature, because of slower growth rates during immature stages [8,9]. An indirect outcome is that insect behaviors and several life-history traits usually associated with size, such as fecundity and longevity, are in turn affected, and those responses to temperature constraints strongly affect fitness [10,11]. As a consequence, temperature plays a major role in terms of ecological patterns that are produced, such as distribution ranges, phenology, and species interactions [1,12].

Insect performance (e.g., locomotion, growth, longevity, reproduction) is commonly bounded between two critical thermal limits (CTLs), beyond which survival is impossible [13]. CTLs are usually assessed by cooling or heating an organism from an initial temperature until physiological or behavioral failure [13,14]. There is no doubt that insects cannot respond optimally to all temperatures, because life-history traits and strategies are linked together in complex trade-offs depending on the experienced thermal regimes [15,16]. Therefore, most organisms have evolved physiological and behavioral mechanisms allowing to cope with sub-optimal temperatures [17], deviating from the thermal optimum at which performance is optimized in a given habitat.

In addition to genetic adaptations, phenotypic plasticity plays a major role in the relatively short-term responses of insects to temperature variations [18,19]. Thermal acclimation is generally defined as the plastic adjustment of life-history traits of an individual in response to changes in temperature [4,20]. Ectothermic organisms acclimated to a given temperature usually show enhanced behavioral and physiological performances at that temperature when compared to other temperatures or to non-acclimated organisms, which is known as the beneficial acclimation hypothesis [21]. Acclimation through extended exposure to a raise of temperature among or across generations causes physiological adjustments that can increase trait performance at higher temperatures and can extend the critical thermal maximum (CTmax) [20,22,23]. For example, in *Drosophila melanogaster*, both acclimation temperature and rate of warming significantly improves CTmax, although the effect is relatively weak for acclimation [24]. In general, CTmax values are weakly plastic to variations in temperature (i.e., not prone to a strong acclimation effect) [25]. However, estimation of the CTmax limit and its variability is particularly relevant to assess the species’ extinction risk to global warming, especially in insects already experiencing temperatures close to their upper tolerance range [26,27].

There are often large variations in temperature among generations or throughout the life span of an individual [28], so considering what occurs during insect ontogenesis in terms of thermal biology (i.e., developmental plasticity) is crucial [29,30,31]. Moreover, in the context of climate change, organisms are being exposed to higher mean temperatures, in addition to increased frequencies of extreme thermal events often exceeding their upper physiological tolerance limit [32,33]. Thermal developmental plasticity may help insects experiencing increasingly high temperatures buffer the negative effects of climate change [19], even if the generality and the role of such beneficial acclimation hypothesis in climate change adaptation of insects is long-time debated [21,34,35]. Therefore, understanding the variety of adaptive mechanisms and plastic responses of insects to different temperatures and studying their thermal tolerance limits are relevant both in theoretical thermal biology, for example, studying insect fitness and population dynamics under thermal stress and in applied ecology for the conservation of insect populations facing climate change or in the context of biological pest control [19,36,37,38,39].

Parasitoid insects are interesting models to explore how experienced temperatures during development can influence adult fitness through their effect on life-history traits [6,7,40], thermal tolerance capacities [41], or the interaction between trait expression and thermal tolerance [42,43]. Acclimation can benefit parasitoids by modulating the negative effects of thermal stress on reproduction or longevity, for example, in case of exposure to cold conditions [44]. Parasitoids are also key models in behavioral ecology, and some studies address the fact that developmental thermal plasticity may affect host-foraging behaviors [44,45]. Indeed, temperature deeply affects the way species interact with each other, and thermal variations may both directly and indirectly alter the outcomes of host-parasitoid interactions [46,47].

In this study, we focused on the aphid parasitoid *Aphidius colemani* Viereck (Hymenoptera: Braconidae), an important natural enemy of aphids used worldwide in biological control strategies. There is a long list of applied and fundamental studies that have focused on this species in terms of basic biology, behavioral ecology, ecophysiology, and its response to temperature variation [48,49,50,51]. Growth rates, body size, and fat mass are affected by rearing temperatures in *A. colemani* [11,52]. By manipulating parasitoid adult size through changes in the rearing temperature, Colinet et al. [11] observed no differences in immature survival rates. However, they showed a negative linear effect of female longevity in relation to developmental temperature, while females developing at mild temperature regimes have a better fecundity. In addition, rearing immature parasitoids at 15 °C improved their cold tolerance, but not their supercooling capacities, as compared to parasitoids reared at 25 °C, probably due to thermal acclimation and complex interaction effects with life-history traits, such as size [52].

Using a multi-trait approach, our goal was to assess how different developmental temperature regimes could affect, as a whole, a set of trait indicators of fitness (egg-load, longevity, mass), physiological state (lipid and water contents), and upper thermal tolerance capacities in *A. colemani*. From an applied perspective, it is relevant to determine what is the appropriate developmental temperature to produce parasitoids that will be released as part of biological control strategies in hot and arid regions, such as in the Maghreb [53]. We hypothesized that parasitoids reared at high temperature regimes would develop the highest heat tolerance capacities, through thermal developmental acclimation. We also expect life-history traits to vary depending on the rearing regime, and as exposed above, either as a result of adaptive plasticity or as a simple response to physiological and metabolic constraints imposed by the temperature.

## 2. Materials and Methods

### 2.1. Biological Material and Acclimation Treatment

The parasitoid *A. colemani* is a polyphagous aphid parasitoid, originating from Northern India or Pakistan, which was introduced in North and South America, Australia, and Europe [54]. *Aphidius colemani* is commercially produced and distributed as an aphid biocontrol agent, targeting primarily *Myzus persicae* and *Aphis gossypii* since 1992 in many European countries [55].

Aphids and parasitoids were obtained from Viridaxis SA (Charleroi, Belgium). *Myzus persicae* were reared on artificial diets [56]. Parasitoids females that were 48 h-old mated and fed were put in the presence of 50 aphid third instar larvae for 3 h at 20 °C. All potentially parasitized aphids were directly transferred either at 10, 20, or 28 °C, under a 16L:8D photoperiod regime and 50% relative humidity level, in climate-controlled rooms. The day they were formed, mummies were individualized in micro-tubes and kept under their respective treatments. Parasitoid adult emergence from the mummies was recorded every 2 h from 8.00 until 20.00, and parasitoids were sexed upon emergence.

### 2.2. Water and Lipid Contents

Newly emerged adults (0–2 h old) were first weighed by using a micro-electronic balance (Me22, Mettler Toledo Inc., Columbus, OH, USA, ±1 µg) to measure fresh mass (FM). They were then dried for 3 days at 60 °C in an air oven and weighed to measure dry mass (DM). Water content was obtained by subtracting DM from FM. Lipids were extracted by placing each dry sample in a centrifugation tube containing 1 mL of the extraction solution (chloroform:methanol, 2:1) for 2 weeks, agitated daily [49]. Adults were then dried again for 12 h in an air oven at 60 °C to remove the extraction solution, and the lean dry mass (LDM) was weighted. Lipid content was obtained by subtracting LDM from DM. The percentages of water and lipids were then calculated by dividing the water and lipid contents by the fresh mass.

### 2.3. Life-History Traits

To estimate the number of mature eggs in the ovary at emergence, 1-h-old females (*N* = 15 for the three developmental temperatures) were dissected in a drop of isotonic saline solution under a binocular (×6.3; Nikon SMZ 800). Mature eggs were distinguished from non-mature eggs, according to Le Ralec [57]. For each female, we measured the length of the hind tibia in the same way as described above.

To assess the longevity of parasitoids, adults (30 males and 30 females per treatment) were isolated individually without host or food in centrifugation tubes (1.5 mL, Eppendorf, Hamburg, Germany). Survival was monitored every day at 20 °C, under a 16L:8D photoperiod regime at 50% relative humidity level until death to determine lifespan.

### 2.4. Critical Thermal Limits

Individuals used for experimentation were 4–12 h old. Insects were heated at 1 °C per minute in a glass column, as used by Powell and Bale [14], from 20 °C to the last critical limit. To control the air temperature in the inner chamber of the column, we used a programmable thermostat bath with circulating water through the outer chamber. Preliminary experiments showed that the temperature was the same throughout the column. During the experiment, the temperature in the inner chamber was recorded using a thermal probe (sensitivity of 0.1 °C). Relative humidity in the tube was constant at 50 ± 10%. For each sex of the three developmental temperatures, two types of response were identified and observed during the heating processes corresponding to two different definitions of CTmax limits found in the literature: (i) CTmax1 was defined as the temperature of onset of muscle spasms [13,58], (ii) CTmax2 corresponds to the heat stupor, i.e., the total paralysis when insect is completely immobile [42,43,59]. The column was raised vertically, and CTmax1 was recorded when the insect had non-coordinated movement and fell at the bottom of the column, and CTmax2 was checked for the last movement.

After measurement of thermal limits, all dead parasitoids were dried at 60 °C for 3 days in an air oven and then weighed individually to measure dry mass using a micro-electronic balance (Me22, Mettler Toledo Inc., Columbus, OH, USA, ±1 µg). We then measured the length of one of the hind tibias. Tibia length is the most commonly used indicator of body size in parasitoid wasps [60]. Tibia pictures were recorded using a camera (Panasonic, Osaka, Japan, Super Dynamic, WV-CP450) mounted on a stereo microscope (×4, Leica MZ6, Wetzlar, Germany), and measures were taken using the ImageJ software (Rasband, W.S., US National Institutes of Health, Bethesda, Rockville, MD, USA). Surface to volume ratio (SVR) was estimated by calculating the ratio of tibia length to dry body mass for each individual tested [42].

### 2.5. Statistical Analyses

The effects of two factors, developmental temperature and sex, and their interaction on the different phenotypic traits were investigated through linear and generalized linear models. Both developmental temperature and sex were analyzed as qualitative variables in the models. Fresh mass, SVR, percentages of water, and lipids, as well as upper thermal limits, were analyzed with linear models, as residuals did not deviate from normality nor homoscedasticity. Tukey’s tests were used for post-hoc comparisons. Egg load at emergence was analyzed through a generalized linear model, assuming a Poisson distribution (log link function) and tibia length was always introduced as the first explanatory variable to remove any effect of animal size [61]. Parametric survival analysis, assuming a non-constant hazard function, following a Weibull distribution, was used to analyze the adult longevity, and the significance of the explanatory variables was assessed using z statistics. Tukey’s tests were used for post-hoc comparisons. Global correlations using R Pearson tests were performed to investigate the link between upper thermal limits and SVR. All analyses were carried out using R software version 4.0.4 [62], and graphics were obtained using the plotCI and ggplot2 packages [63].

## 3. Results

### 3.1. Morphology and Physiology

Fresh mass of parasitoids differed significantly with the developmental temperature (F_2,172_ = 219.28, *p* < 0.0001; Figure 1a). Their fresh mass decreased with the developmental temperature and was maximal when they developed at 10 °C. Furthermore, the fresh mass of males and females tended to respond similarly to the developmental temperature (F_2,172_ = 2.67, *p* = 0.071). The surface to volume ratio (SVR) did not differ between males and females (F_1,169_ = 0.10, *p* = 0.75) and increased with the developmental temperature (F_2,169_ = 19.38, *p* < 0.0001; Figure 1b).

The developmental temperature and the sex had an interactive effect on the percentage of water (F_2,168_ = 4.14, *p* = 0.018; Figure 1c). Parasitoids developed at 20 °C showed the lowest water percentage, and males tended to present a lower percentage of water than females when they developed at 20 °C and 28 °C (Tukey pairwise comparisons: at 20 °C, *p* = 0.027; at 28 °C, *p* = 0.037). In contrast, the lipid percentage was affected by developmental temperature (F_2,159_ = 51.94, *p* < 0.0001) but not by the sex (F_1,159_ = 2.43, *p* = 0.121). Parasitoids accumulated more lipids when they developed at 20 °C than at 10 °C and 28 °C (Figure 1d).

### 3.2. Fitness-Related Traits

Parasitoid size positively influenced egg load at emergence (χ^2^_1_ = 70.24, *p* < 0.0001). After removing the body size effect, egg load at emergence was not influenced by the developmental temperature (χ^2^ = 0.3, *p* = 0.86; Figure 2a). Developmental temperature had a significant effect on adult survival, with parasitoids developed at 20 °C showing a better survival than those developed at 10 °C and 28 °C (z = 2.8, *p* = 0.005; Figure 2b). However, adult survival was not different between male and female parasitoids (z = −0.47, *p* = 0.64).

### 3.3. Critical Thermal Limits

Critical thermal limits for heat stupor (CTmax1) and heat coma (CTmax2) were influenced by the developmental temperature (CTmax1: F_2,175_ = 115.52, *p* < 0.0001; CTmax2: F_2,174_ = 7.55, *p* = 0.0007). Parasitoids developed at 20 °C showed the highest CTmax1 (post-hoc Tukey tests: *p* < 0.0001; Figure 3a), whereas parasitoids developed at 28 °C showed the highest CTmax2 (post-hoc Tukey tests: 10 °C versus 20 °C: *p* = 0.45; 28 °C versus 20 °C: *p* = 0.0008; 10 °C versus 28 °C: *p* = 0.032). Furthermore, males had a higher CTmax2 than females (F_1,174_ = 6.5, *p* = 0.012; Figure 3b). Neither CTmax1 nor CTmax2 were significantly correlated to SVR (CTmax1: r = 0.12, *p* = 0.11; CTmax2: r = 0.14, *p* = 0.07). Furthermore, CTmax1 and CTmax2 were not correlated (r = 0.13, *p* = 0.075).

## 4. Discussion

Several studies have previously underlined the effect of temperature on the aphid parasitoid *A. colemani* in terms of physiology, morphology, or behavior [11,51,53,56]. In this study, we confirmed that developmental temperature could affect various phenotypic traits of adult *A. colemani*, such as the physiological state, fitness-related traits, and upper thermal tolerance capacities. Our results have potential applied implications to improve biological control programs in arid regions, such as in the Maghreb, where temperatures at night can drop to 20 °C but raise above 30 °C during the day.

Developmental temperature has important and various effects on phenotypic traits of parasitoids [7,36,61]. The effect of developmental temperature on size and fresh mass confirmed previous results on *A. colemani* [11]. More generally, the observed larger size of parasitoids after development at low temperature is in accordance with the temperature-size rule reported in ectotherms [9]. Due to allometric scaling, the effect of temperature on insect body size has important consequences on other phenotypic traits [64,65,66]. In this study, we observed that the variation in egg load at emergence, according to developmental temperature, was completely explained by the variation in size. However, developmental temperature can also impact phenotypic traits independently of its effect on body size. Thus, we observed that water and lipid contents relative to fresh mass varied with the developmental temperature and were minimal and maximal when parasitoids developed at 20 °C, respectively. This result is in contrast with the linear decrease of fat content with rearing temperature that has already been reported in a mummy of *A. colemani* [11]. Our study suggests that the variation of lipid content can be described by the typical model of thermal performance curve [18], with a maximal accumulation of lipids at an optimal developmental temperature (20 °C). A similar pattern has been observed in the parasitoid *Venturia canescens,* in which thelytokous parasitoids emerged with a larger lipid content after development at constant intermediate temperature than after development at colder or hotter temperatures [67].

The fat reserve was previously considered as a non-replaceable resource in parasitoids [68,69]. If it needs to be re-evaluated [70], lipid contents remain positively correlated with several fitness-related traits, such as longevity, dispersal, and reproduction [65,71]. In accordance, parasitoids reared at 20 °C emerged with a larger lipid content and survived longer when maintained at 20 °C than parasitoids developed at 10 °C and 28 °C. It is possible that females developing at 10 °C would become time-limited, as their metabolic rate would be accelerated once placed at 20 °C, following the Thermal Compensation Hypothesis (TCH), which states that metabolic rates of cold-acclimated species increase once exposed to relatively higher temperatures, compared to warm-adapted species [72]. According to Le Lann et al. [43], higher metabolic rates may result in a modification of the energy allocation between longevity and other traits or functions. Therefore, it is important to consider trade-offs that can occur among traits, such as between longevity and egg-load. Finally, parasitoids reared at 28 °C could also be energy-limited because of high metabolic rates or exposure to higher desiccation stress during immature development, which can reduce their survival probability at the adult stage.

The development at high temperature led to a weak improvement of the upper thermal limits to activity. Indeed, the critical thermal limits for a heat coma (CTmax2) was slightly higher in parasitoids developed at the highest temperature, whereas the critical thermal limits for a heat stupor (CTmax1) was maximized after development at intermediate temperature. We conclude that there is only weak support for the beneficial acclimation hypothesis in this species, regarding heat tolerance. The lack of strong acclimation effects on the upper thermal limits was underlined on other insect species [73,74,75]. These results point to the limits of phenotypic plasticity on thermal tolerance, especially on the upper bound of the critical thermal limits. We also could not reveal any correlation between the SVR and the maximal thermal limits. There is a striking lack of data in the literature regarding this potential correlation, but most of the time, no correlation is shown, suggesting that the CTmax is generally independent from the size, biomass, and water content [76].

In an applied context of biological pest control in an arid area, it is important to find an adequate solution to help parasitoids sustain the detrimental effects of high temperature. Parasitoid efficiency on the control of pest populations in regions with above-optimal temperatures depend on the extent to which life-history traits are adapted to these thermal conditions, because these traits can, in turn, influence survival probabilities and optimal decision-making by females exploiting host patches [77,78]. Our results show that developmental temperatures are crucial to determine how biological control strategies can be improved, especially because commercial strains of parasitoid are often reared under constant 20 °C condition, while released parasitoids experience various thermal conditions. Next steps should be to look for genetic variations in thermal tolerance plasticity in different populations of aphid parasitoids to select strains most likely to fit a target thermal environment. A particular focus should also be made on the effect of thermal fluctuations during development, which could greatly influence adult responses and would, in any case, be more relevant to consider from an ecological point of view than simple constant regimes [79].

## Figures and Tables

**Figure 1 insects-12-00852-f001:**
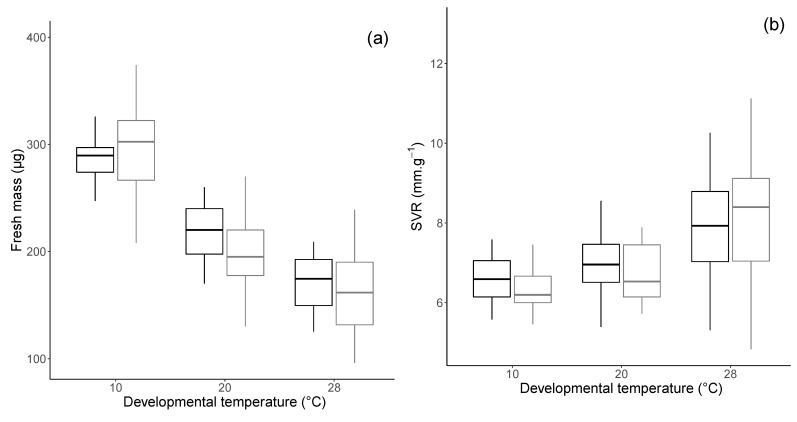
Fresh mass (**a**), surface/volume ratio SVR (**b**), percentage of water (**c**), and lipids (**d**) of female (black) and male (grey) *A. colemani*, according to developmental temperature (10, 20, or 28 °C). Percentage of water and lipids correspond to contents divided by parasitoid fresh mass. Bold line: median; box: lower and upper quartiles; whiskers: smallest and largest non-outlier observations.

**Figure 2 insects-12-00852-f002:**
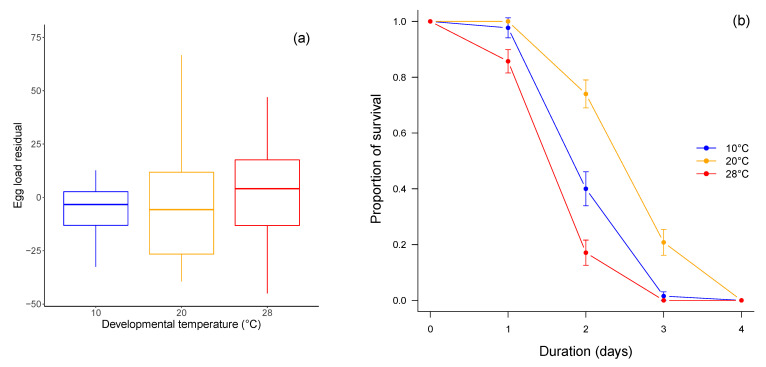
Egg load at emergence (**a**) and adult survival (mean ± SE) at 20 °C (**b**) of parasitoids developed at 10 °C (blue), 20 °C (orange), and 28 °C (red). Egg load at emergence is size corrected. In the boxplots, bold line: median; box: lower and upper quartiles; whiskers: smallest and largest non-outlier observations. For ease of reading and due to the absence of significant difference, male and female survivals were combined.

**Figure 3 insects-12-00852-f003:**
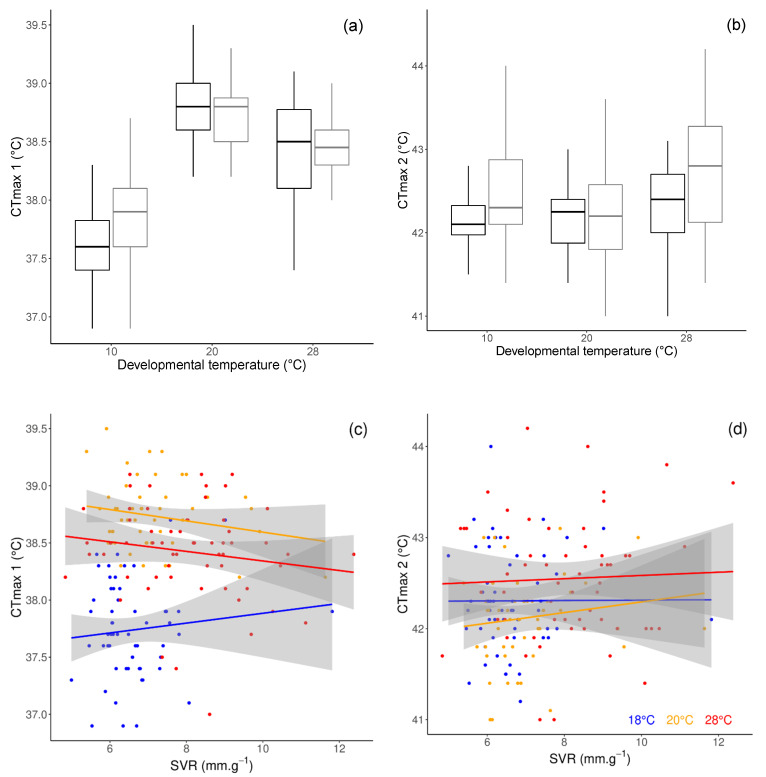
Critical thermal limit for heat stupor (CTmax1) (**a**) and heat coma (CTmax2) (**b**) of female (black) and male (grey) *A. colemani*, according to developmental temperature. Bold line: median; box: lower and upper quartiles; whiskers: smallest and largest non-outlier observations. Correlation between surface to volume ratio (SVR) with CTmax1 (**c**) and CTmax2 (**d**). For each developmental temperature, a linear regression smoothing was performed over the individual data (blue: 18 °C, orange: 20 °C, red: 28 °C), and grey areas correspond to confidence intervals around the regressions.

## Data Availability

Analyses reported in this manuscript can be reproduced using the data available in the free repository server Zenodo: https://doi.org/10.5281/zenodo.5520999.

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
