# Peer review of "Developmental Temperature Affects Life-History Traits and Heat Tolerance in the Aphid Parasitoid Aphidius colemani"

_insects, 2021, doi:10.3390/insects12100852_

Round 1
Reviewer 1 Report
This article by Jerbi-Elayed and colleagues (insects-1380922) is well motivated, the structure is appropriate, and the manuscript is well written without missing any key details. The methods used are appropriate for the objectives of the work and, in general, well depicted. The resulting figures are sufficient, informative, and of good quality helping to follow the reasoning throughout the manuscript. The discussion of results and comments on future research was nicely done and will be useful to others. Overall, I enjoyed reading the manuscript.
A few editorial remarks have been made below for authors to consider.
Some of the authors statements would be much stronger if they tie their work to the body of literature that has built up on the bioecology of endo and ectoparasites biocontrol agents across the range of realistic temperatures experienced in the field. They all point to the same direction. Some examples are J. Econ. Entomol. 112: 1560-1574 (ectoparasites biocontrol agents) or J. Econ. Entomol. 112:1062-1072 (endoparasites biocontrol agents), but there are others. These studies provide strong evidence of increased longevity in biocontrol agents reared at non-stressful low temperatures when compared to higher temperature regimes. This article should provide details on all these fronts to provide the proper context for the work, e.g., on lines 49 through 41, 58 through 60, 97 through 99, and 102 through 105 in your text. They also indicate that the parasitism or egg load was significantly higher at intermediate temperatures than at cline margins, e.g., on lines 260 through 261 and 282 through 284 in your text (statements are missing reference). Adding these details will improve the paper.
Good luck!
Author Response
Response to Reviewer 1 Comments
Reviewer 1
This article by Jerbi-Elayed and colleagues (insects-1380922) is well motivated, the structure is appropriate, and the manuscript is well written without missing any key details. The methods used are appropriate for the objectives of the work and, in general, well depicted. The resulting figures are sufficient, informative, and of good quality helping to follow the reasoning throughout the manuscript. The discussion of results and comments on future research was nicely done and will be useful to others. Overall, I enjoyed reading the manuscript.
A few editorial remarks have been made below for authors to consider.
Some of the authors statements would be much stronger if they tie their work to the body of literature that has built up on the bioecology of endo and ectoparasites biocontrol agents across the range of realistic temperatures experienced in the field. They all point to the same direction. Some examples are J. Econ. Entomol. 112: 1560-1574 (ectoparasites biocontrol agents) or J. Econ. Entomol. 112:1062-1072 (endoparasites biocontrol agents), but there are others. These studies provide strong evidence of increased longevity in biocontrol agents reared at non-stressful low temperatures when compared to higher temperature regimes. This article should provide details on all these fronts to provide the proper context for the work, e.g., on lines 49 through 41, 58 through 60, 97 through 99, and 102 through 105 in your text. They also indicate that the parasitism or egg load was significantly higher at intermediate temperatures than at cline margins, e.g., on lines 260 through 261 and 282 through 284 in your text (statements are missing reference). Adding these details will improve the paper.
The first paragraph of the introduction has been rephrased to include the suggested references (Lines 49-51). Further, these references have been included to illustrate the important effect of developmental temperature on fitness related traits of biological agents (Lines 106 & 266).
Good luck!
Thank you !
Reviewer 2 Report
Temperature is one of the most important factors influencing insect physiology and behavior. Therefore, any new research devoted to thermal effects could be interesting. The authors of the present manuscript investigated the influence of temperature conditions of development on several important biological parameters (in particular, heat tolerance) of adults of Aphidius colemani, an insect parasitoid widely used for biological control of aphids. Although a number of similar studies were conducted with this and other insect species, the present study yielded some new and interesting results that can be important for both fundamental and applied entomology. All experiments were well planned and conducted; the results were properly analyzed; all conclusions are supported by the data. Thus, the manuscript deserves publication, although some (minor) revision is still necessary (see my comments below).
Line 32: I would suggest replacing of ‘show’ by ‘showed’ because all other results are described in the Abstract in the past tense.
Lines 143 and 167: for uniformity, I would suggest using of one and the same format for photoperiod description.
Lines 148 and 169: it was noted above (line 146) that adult emergence was recorded twice a day with intervals of 10 h (day) and 14 h (night). If so, how the age groups of 0–2 h (line 148) and 4-12 h (line 169) were selected?
Line 195: I guess, ‘temperature’ should be inserted after ‘developmental’.
Line 225: please, indicate in the legends what is shown in Figs. 1a-d, 2a and 3a-b: median, quartiles and range or something else?
Line 281: please, replace year with serial number of publication, especially since in the list of references there are two papers published by Le Lann and co-authors in 2011 (see lines 408-411).
Line 486: I do not know what is the meaning of “n/a-n/a” in this case, but the full correct citation of this paper is: Ecological Entomology, 38, 355–363.
Author Response
Response to Reviewer 2 Comments
Reviewer 2
Temperature is one of the most important factors influencing insect physiology and behavior. Therefore, any new research devoted to thermal effects could be interesting. The authors of the present manuscript investigated the influence of temperature conditions of development on several important biological parameters (in particular, heat tolerance) of adults of Aphidius colemani, an insect parasitoid widely used for biological control of aphids. Although a number of similar studies were conducted with this and other insect species, the present study yielded some new and interesting results that can be important for both fundamental and applied entomology. All experiments were well planned and conducted; the results were properly analyzed; all conclusions are supported by the data. Thus, the manuscript deserves publication, although some (minor) revision is still necessary (see my comments below).
Line 32: I would suggest replacing of ‘show’ by ‘showed’ because all other results are described in the Abstract in the past tense.
Done
Lines 143 and 167: for uniformity, I would suggest using of one and the same format for photoperiod description.
Done
Lines 148 and 169: it was noted above (line 146) that adult emergence was recorded twice a day with intervals of 10 h (day) and 14 h (night). If so, how the age groups of 0–2 h (line 148) and 4-12 h (line 169) were selected?
Indeed, there is a mistake in the first version of the manuscript. Parasitoid emergence was recorded every two hours from 8:00 until 20:00. We corrected the new version accordingly (Line 151)
Line 195: I guess, ‘temperature’ should be inserted after ‘developmental’.
Done
Line 225: please, indicate in the legends what is shown in Figs. 1a-d, 2a and 3a-b: median, quartiles and range or something else?
We have added these indications in the legend of the figures
Line 281: please, replace year with serial number of publication, especially since in the list of references there are two papers published by Le Lann and co-authors in 2011 (see lines 408-411).
Done
Line 486: I do not know what is the meaning of “n/a-n/a” in this case, but the full correct citation of this paper is: Ecological Entomology, 38, 355–363.
The correct citation has been added.